# Robust Two-Hand Reconstruction with Additional 2D Information and Diffusion Prior Conference Submissions

## Abstract

Recently, estimating 3D hand pose and shape from monocular images has garnered significant attention from researchers, which finds numerous applications in animation, AR/VR, and embodied AI. Many tasks in the field of computer vision have demonstrated the substantial benefits of incorporating additional task-relevant reference information to enhance model performance. In this paper, we investigate whether the principle of "the more you know, the better you understand" also applies to the task of two-hand recovery. Unlike previous methods that rely solely on monocular image features for hand estimation, we extract 2D keypoints, segmentation map, and depth map features and then integrate them with image features. The hand regressor subsequently estimates hand parameters based on the fused features. The 2D keypoints and segmentation maps provide detailed finger XY-dimensional reference information for the hand, while the depth map offers pixel-level relative Y-dimensional reference information. Recovering the 3D hand from these intermediate representations should be more straightforward than doing so solely from the original RGB image. Current foundation models have already achieved impressive performance on these basic tasks, allowing us to obtain reliable results in most cases. However, when the two hands overlap significantly, resulting in complex entanglements. In such cases, hand penetration is likely to arise. The additional reference information (segmentation map and depth map) cannot assist with the occluded regions, and the predicted 2D keypoints for the occluded areas are also unreliable. To this end, we further employ a two-hand diffusion model as a prior and employ gradient guidance to refine the two-hand contact. Extensive experiments demonstrate that our approach achieves superior performance in 2D consistency alignment and depth recovery.

## 1 Introduction

3D two-hand recovery aims to reconstruct both hands of a person in 3D space, a crucial task for numerous emerging applications, including 3D character animation, augmented and virtual reality (AR/VR) and robotics. Large-scale hand datasets (Moon et al., 2020; 2024) have greatly facilitated numerous studies on hand recovery. These approaches can be summarized as focusing on scaling up datasets (Pavlakos et al., 2024), improving backbone (Lin et al., 2021; Pavlakos et al., 2024), and incorporating attention mechanisms (Li et al., 2022; Yu et al., 2023; Lin et al., 2024) between the hands.

Recent developments in computer vision point towards a trend where advances are driven by the incorporation of additional task-relevant reference information. For instance, ECON (Xiu et al., 2023) introduces rendered front and back body normal images as input for human digitization, enabling the model to effectively infer high-fidelity 3D humans in loose clothing and challenging poses. In the task of 3D human motion recovery, WHAM (Shin et al., 2024) utilizes 2D human keypoints to extract motion features as inputs, leading to more robust and stable 3D human motion estimation. More additional detailed input allows the model to gain a deeper context understanding of the task while minimizing uncertainty and ambiguity. Currently, whether integrating additional informative cues can enhance model performance in the domain of hand recovery remains unexplored.

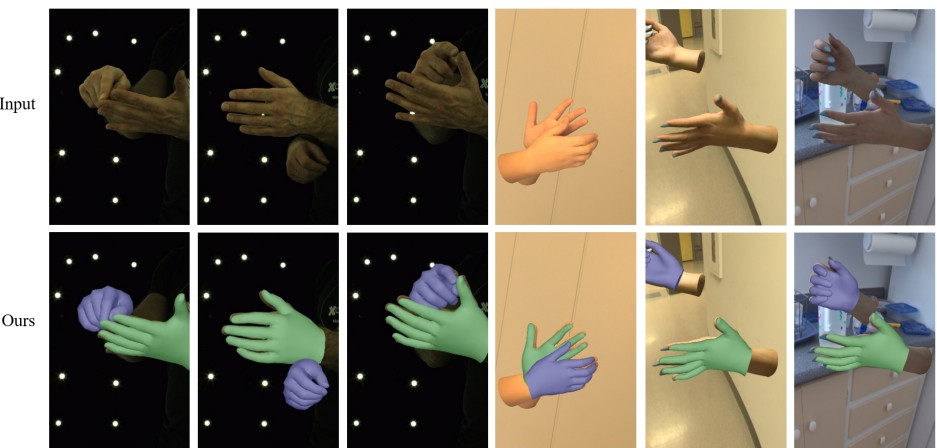

Figure 1: 3D two-hand recovery in InterHand2.6M and Re:InterHand test datasets.

In this paper, we adopt the philosophy "the more you know, the better you understand" and apply it to the problem of 3D two-hand estimation. We primarily consider incorporating 2D keypoints, segmentation maps, and depth maps as additional reference inputs. The rationale behind this is that 2D keypoints provide precise spatial information about hand joints in the XY plane, segmentation maps offer a detailed understanding of hand shape and boundaries, while depth maps supply pixel-level depth cues, helping to better capture the hand's relative positioning in 3D space. Specifically, we leverage the foundation model to extract 2D keypoints, segmentation maps, and depth maps from monocular images. Then, using neural networks extract features from these intermediate information and fuse them with image features. The hand regressor subsequently estimates hand parameters based on the fused features.

Although the additional 2D information has provided more detailed references for hand estimation models, we consider a special case where one hand occludes important fingers of the other hand, leading to penetration issues between the estimated hands. In such cases, the three types of additional information mentioned above are ineffective in providing valid references for the occluded parts. The diffusion model has achieved considerable success across various fields and has been demonstrated to effectively learn the data distribution as a prior. In the field of human estimation, BUDDI (Müller et al., 2024) reconstructs two individuals in close proximity by utilizing the diffusion model as a prior. Similarly, DPoser (Lu et al., 2023) constructs a robust human pose diffusion model for pose generation, pose completion, and motion denoising. Inspired by this, we use a pretrained two-hand diffusion model (Lee et al., 2024) as the prior to address hand penetration issues. Given an initial regression two-hand estimate, we invert it to the corresponding noise. Then, we propose a penetration gradient guidance during the denoising process.

Extensive ablation experiments further validate the improvements in model performance brought by the incorporation of additional 2D information and the use of diffusion. Our qualitative experiments in real-world scenarios also demonstrate superior alignment and depth recovery capabilities. As shown in Figure 1, we achieve robust hand reconstruction across various scenes and interactive poses.

Our key contributions can be summarized as follows.

- We propose incorporating additional 2D reference information for the 3D hand recovery task, including 2D keypoints, segmentation maps, and depth maps. Specifically, after aligning with the image space using a visual backbone model, we employ a simple transformer encoder to integrate these features.

- When one hand is occluded by another, leading to potential hand penetration issues. The additional 2D reference information does not assist with the occluded regions. In this context, we propose utilizing a pretrained diffusion model as a prior and employing gradient guidance to address the hand penetration issue.

## 2 RELATED WORK

### 2.1 3D HAND RECOVERY

With the introduction of some high-quality hand datasets, recovering 3D hand MANO (Romero et al., 2017) parameters from monocular input images has recently achieved remarkable advancements. For single-hand recovery, METRO (Zhang et al., 2019) employs a convolutional neural network to extract a single global image feature and performs position encoding by repeatedly concatenating this image feature with the 3D coordinates of a mesh template. MeshGraphormer (Lin et al., 2021) introduces a graph-convolution enhanced transformer to effectively model both local and global interactions. AMVUR (Jiang et al., 2023) proposes a probabilistic approach to estimate the prior probability distribution of hand joints and vertices. Zhou et al. (Zhou et al., 2024) simplifies the process by decomposing the mesh decoder into a token generator and a mesh regressor, achieving high performance and real-time efficiency through a straightforward yet effective baseline. HaMeR (Pavlakos et al., 2024) highlights the significant impact of scaling up to large-scale training data and utilizing high-capacity deep architectures for improving the accuracy and effectiveness of hand mesh recovery. For two-hand recovery, IntagHand (Li et al., 2022) propose a GCN-based network to reconstruct two interacting hands from a single RGB image, featuring pyramid image feature attention (PIFA) and cross hand attention (CHA) modules to address occlusion and interaction challenges. InterWild (Moon, 2023) bridges MoCap and ITW samples for robust 3D interacting hands recovery in the wild by leveraging single-hand ITW data for 2D scale space alignment and using geometric features for appearance-invariant space. ACR (Yu et al., 2023) explicitly mitigates interdependencies between hands and between parts by leveraging center and part-based attention for feature extraction. 4DHands (Lin et al., 2024) handles both single-hand and two-hand inputs while leveraging relative hand positions using a transformer-based architecture with Relation-aware Two-Hand Tokenization (RAT) and a Spatio-temporal Interaction Reasoning (SIR) module. Although these methods have generally achieved good results in hand pose and shape reconstruction, their performance in finer details is still lacking. Following the adage "the more you know, the better you understand," we explore the integration of additional 2D information to guide 3D hand recovery.

### 2.2 INTEGRATING ADDITIONAL INFORMATION

Recently, many studies have attempted to introduce additional reference information as guidance in visual tasks to achieve better performance. For example, in the task of human digitization, ECON (Xiu et al., 2023) takes as input an RGB image and is conditioned on the rendered front and back body normal images. This strategy allows it to excel at inferring high-fidelity 3D humans in loose clothing and challenging poses. For text-to-image generation, ControlNet (Zhang et al., 2023) has also successfully utilized different types of conditional inputs, such as sketches, depth maps, and segmentation maps. It has successfully achieved the generation of images aligned with these conditional guides using a pretrained text-to-image diffusion model. These impressive results demonstrate the powerful capability of neural networks to fit guiding information. For the 3D human motion estimation task, WHAM (Shin et al., 2024) uses human 2D keypoints to extract motion features as inputs for both the Motion Decoder and Trajectory Decoder. This approach achieves more robust and stable 3D human motion estimates in global coordinates. Currently, whether incorporating additional guiding information can enhance hand recovery performance remains unexplored. We attempt to use a foundation model (Khirodkar et al., 2024) to obtain intermediate 2D hand information, including keypoints, segmentation maps, and depth maps, to achieve more robust hand estimation.

## 3 METHOD

In this Section, we present the technical details of our method. As illustrated in Fig. 2, distinguishing from previous hand estimation methods, our approach primarily involves introducing additional 2D information as input to guide the two-hand MANO (Romero et al., 2017) parameter estimation. The estimated two-hand parameters are then refined using a two-hand diffusion. Therefore, our method can be divided into two stages: A two-hand estimator followed by two-hand diffusion model. In most cases, the two-hand estimator from the first stage is capable of accurately fitting the hand parameters by incorporating additional 2D information. However, when the Intersection over Union

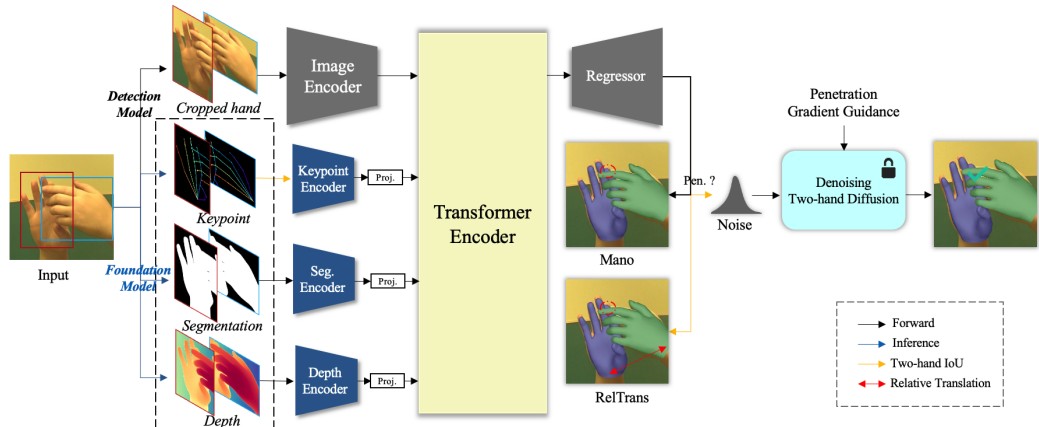

Figure 2: The overall pipeline of our proposed method. "Seg.", "Pen." and "RelTrans" are abbreviations for "Segmentation", "Penetration" and "Relative Translation", respectively.

(IoU) between the two hands approaches one, one hand can severely occlude another, resulting in penetration problem and 2D information cannot provide assiatance for occluded part. In such scenarios, employing the second stage involving two-hand diffusion proves to be very beneficial.

## 3.1 HAND RECOVERY WITH ADDITIONAL INFORMATION

Previous methods take a monocular image as input and utilize a backbone network to extract image features for the regression of hand MANO parameters. We explore incorporating additional information to enhance the regressor's ability to fit two-hand parameters more effectively. Given that the hand recovery task involves reconstructing a 3D hand model, clean and direct 2D information may facilitate the model's understanding and fitting. Therefore, we consider using 2D keypoints, segmentation maps, and depth maps. Specifically, we input the monocular image into the human foundation model Sapiens (Khirodkar et al., 2024) to obtain these results. In the following subsections, we analyze these types of information in detail.

**2D Keypoints.** 2D keypoints provide precise locations of important hand features (like joints and fingertips), enabling better understanding of hand poses. They reduce the complexity of the data by focusing on key points rather than the entire image, making feature extraction more efficient. Accurate estimation of 2D keypoints for both hands remains a challenge for current baseline models when there is significant hand overlap. Therefore, to ensure stability during inference, we reduce reliance on keypoint information when the IoU between the hands exceeds 0. In such cases, we use only the wrist points for hand positioning. As shown in our pipeline, the yellow arrow indicates the IoU threshold check. To align for feature fusion in image space, we first visualize the keypoint data as an RGB image, and then employ a visual backbone network to extract the features of the hand's 2D keypoints. The visual backbone network also provides pre-trained weights that facilitate effective training for extracting keypoint features.

**Segmentation map.** Segmentation maps provide pixel-level information, allowing for precise localization of hand and its parts. They also help isolate hands from the background, reducing noise and distractions. By focusing on segmented areas, models can extract relevant hand features more effectively. It is worth noting that when there is significant interleaving of the two hands, the prediction results for the 2D keypoint pairs of the hands can be quite unreliable. However, the segmentation map can still provide reasonably accurate 2D contour information of the hands at this time (the hand segmentation map does not distinguish between the left and right hands when the Intersection over Union (IoU) for both hands is greater than 0). We also use a visual backbone network to extract the features of the segmentation map.

**Depth map.** Depth maps provide information about the distance between the hands and the camera, helping to capture the relative positioning and spatial relationship of the hands in a real environment.

Depth information is less affected by variations in lighting conditions, making hand understanding more reliable in environments with varying or poor lighting. We also use a visual backbone network to extract the features of the depth map.

**Multiple features fusion with transformer encoder.** We consider using a simple approach to fuse different feature information, specifically by employing a Transformer encoder. The self-attention mechanism in the Transformer encoder allows each feature token to attend to every other token in the input sequence. This enables the model to capture long-range dependencies and integrate information globally, rather than being limited to local context as in traditional models like RNNs or CNNs. Given $I_i$, $I_k$, $I_m$, and $I_d$ to represent the input of hand image, 2D keypoints, segmentation map and depth map, the fused feature $F$ can be expressed as:

$$F = TransEnc(< (f^i(I_i)), f^k(I_k), f^s(I_s), f^d(I_d) >)[0:s], \tag{1}$$

Here, $<,>$ denotes the concat operation. $TransEnc$ represents the Transformer encoder. $f^i$ represents the feature extraction network for images, while $f^k$, $f^s$, $f^d$ denotes the networks for extracting features from 2D keypoints, segmentation maps, and depth maps, respectively. Each of these networks consists of a smaller model compared to the image extraction network, followed by a projection layer. $s$ denotes the image feature length.

**Loss function.** The fused feature $F$ is then input into the two-hand regressor to regress the parameters of both hands and relative translation. Building on previous two-hand recovery approaches (Moon, 2023; Yu et al., 2023), We train our model in an end-to-end fashion by minimizing the L1 distance between the predicted and ground truth (GT) MANO parameters, the 3D and 2.5D joint coordinates, as well as the 3D relative translation.

### 3.2 TWO-HAND REFINING WITH TWO-HAND DIFFUSION MODEL

Although the additional information discussed in the previous section provides more detailed references for the hand estimation model, we still consider a special case where one hand occludes the important fingers of the other hand. In this scenario, the three types of additional information mentioned earlier are unable to provide effective guidance for the occluded parts of the hand. Consequently, the estimated occluded regions of both hands are prone to penetration issues. To address this, we introduce a pre-trained two-hand diffusion model as a prior to adjust the predicted results, using the penetration gradient as guidance. we employ the unconditional version of the two-hand generation diffusion model from (Lee et al., 2024), which denoise hand noise into MANO parameters. As shown in Fig. 2, before using the two-hand diffusion model, we perform an IoU and penetration check between the two hands to reduce unnecessary diffusion inference in most cases. The gradient-guided two-hand diffusion effectively alleviates the penetration problem of the occluded regions.

**Two-hand penetration refining with gradient guidance.** Given an estimated two hand result, we first invert it to the corresponding noise. We then introduce a gradient-guided strategy to prevent hand penetration. The two-hand diffusion model operates in a cascaded manner, first denoising one hand and then using the denoised hand as a condition to denoise and generate the second hand. Since the visible hand is often reconstructed more accurately, we use it as the conditional hand for mitigating interpenetration for another hand. We calculate the collision loss between it and the occluded hand from each step of reverse diffusion and guide the occluded hand to move in the direction of the negative gradient. Specifically, during each denoising step, we generate clean occluded hand parameters $\hat{X}_0$ from the current noisy occluded hand $X_{t-1}$ through the DDIM sampling process. These parameters, along with the reference hand for interpenetration, are then input into the MANO model to obtain the mesh vertices $V_{t-1}$ and $V_c \in \mathbb{R}^{778 \times 3}$. The formula is as follows:

$$V = MANO(\sqrt{\alpha_{t-1}}\hat{x}_0 + \sqrt{1 - \alpha_{t-1} - \sigma_t^2} \cdot \epsilon_t^\theta(x_t)), \tag{2}$$

where $\alpha_t = \prod_{s=1}^t (1 - \beta_s)$ and $\{\beta_t\}_{t=1}^T$ is the variance schedule. Subsequently, we introduce a collision detection function $C_{col}$ to detect the indices of vertices where collisions occur between the occluded hand and the reference hand. Specifically, it first calculates the Chamfer distance between all pairs of vertices and selects the nearest index pair $N_i$ for each vertex. Then, it computes the angle between their normal vectors to select valid collision points (we use a threshold of 1 radian). We

define the collision check formula as follows:

$$N_i = \sum_i \sum_j argmin_j(D_{ij}(V_{t-1}^i - V_c^j)^2).$$

$$C_{col} = (i, j) if cos(\theta_{ij}) < cos(\theta_{threshold}),$$

(3)

Finally, we introduce the GMoF function to process the collision loss $L_{collision}$, making it more robust to outliers and noise. The result is then activated as a gradient and applied in the negative direction to $\hat{X}_0$:

$$L_{collision} = \sum_i \sum_j (\frac{||V_{t-1}^i - V_c^j||^2}{||V_{t-1}^i - V_c^j||^2 - \rho}),$$

$$\hat{X}_0 = \hat{X}_0 - \lambda(\delta_i L_{collision}),$$

(4)

where $\rho$ is set to 5e-2, and $\lambda$ is the weight of the negative gradient direction.

## 4 EXPERIMENTS

### 4.1 IMPLEMENTATION DETAILS

We implement our network using PyTorch (Paszke et al., 2019). For the image feature extractor, we use ResNet-50 (He et al., 2016) as the backbone, while for additional 2d information extractors, we use ResNet-18. Our model is trained on 4 A100 GPUs using the AdamW optimizer, starting with an initial learning rate of 1e-4, which is reduced by a factor of 10 at the 4th epoch. We use a mini-batch size of 48. Data augmentations such as scaling, rotation, random horizontal flip, and color jittering are applied during training. The resolution of cropped hand image is $256 \times 256$. During inference, the hand bounding box detector utilizes RTMDet (Lyu et al., 2022). For other details, we follow the approach in (Moon, 2023), where the hand bounding boxes are enlarged by a factor of two to ensure that the entire hand image is captured for feature extraction. The left hand is flipped to align with the right-hand input to the network, and after prediction, the results are flipped back. The 3D relative distance between the hands is predicted based on a 2.5D image. Our training dataset includes InterHand 2.6M (Moon et al., 2020), Re:InterHand (Moon et al., 2024), COCO whole-body (Jin et al., 2020), FreiHand (Zimmermann et al., 2019) and HO-3D (Hampali et al., 2020). For testing, we primarily use InterHand 2.6M, FreiHand and the in-the-wild dataset HIC (Tzionas et al., 2016).

### 4.2 DATASETS

**InterHand 2.6M (Moon et al., 2020)** features both precise human (H) and machine (M) 3D pose and mesh annotations, encompassing 1.36 million frames for training and 850,000 frames for testing. **Re:InterHand (Moon et al., 2024)** is created by rendering 3D hands with precisely tracked 3D poses and applying various environment maps for relighting. The dataset consists of 739K video-based images and 493K frame-based images from third-person viewpoints, and 147K video-based images from egocentric viewpoints. **COCO WholeBody (Jin et al., 2020)** extends the COCO dataset (Lin et al., 2014) by adding comprehensive whole-body annotations. It includes manual annotations covering the entire human body. **FreiHand (Zimmermann et al., 2019)** is a dataset designed for single-hand 3D pose estimation, providing MANO annotations for each frame. It includes $4 \times 32,560$ frames for training and 3,960 frames for evaluation and testing. **HO-3D (Hampali et al., 2020)** focus on hand-object interactions, comprising 66,000 training images and 11,000 test images across 68 different sequences. **HIC (Tzionas et al., 2016)** offers a variety of hand-hand and object-hand interaction sequences, along with 3D ground truth meshes for both hands. We do not use HIC during training. Consequently, we believe that testing on HIC demonstrates the model's generalization ability.

### 4.3 EVALUATION METRICS

We mainly adopt Mean Per Joint Position Error (MPJPE) and Mean Per Vertex Position Error (MPVPE) to measure the 3D errors (in millimeters) of the pose and shape of each estimated hand after aligning them using a root joint translation, and Mean Relative-Root Position Error (MRRPE)

Table 1: Comparison with state-of-the-art on InterHand2.6M(Moon et al., 2020) 5fps test dataset.

| Methods | MRRPE | MPJPE | MPVPE | IH MPJPE | IH MPVPE | SH MPJPE | SH MPVPE |
|---|---|---|---|---|---|---|---|
| Moon et al. (Moon et al., 2020) | - | 13.98 | - | 16.02 | - | 12.16 | - |
| Zhang et al. (Zhang et al., 2021) | - | 11.58 | 12.04 | 11.28 | 12.01 | 11.73 | 12.06 |
| IntagHand (Li et al., 2022) | - | 9.95 | 10.29 | 10.27 | 10.53 | 9.67 | 9.91 |
| ACR (Yu et al., 2023) | - | 8.09 | 8.29 | 9.08 | 9.31 | 6.85 | 7.01 |
| InterWild (Moon, 2023) | 26.74 | 7.85 | 8.16 | 8.24 | 8.68 | 6.72 | 6.93 |
| Ren et.al (Ren et al., 2023) | 28.98 | 7.51 | 7.72 | - | - | - | - |
| 4DHands (Lin et al., 2024) | 24.58 | 7.49 | 7.72 | - | - | - | - |
| **Ours** | **21.80** | **5.41** | **5.67** | **5.97** | **5.91** | **4.85** | **4.88** |

to measure the performance of relative positions (in millimeters) of two hands. Procrustes-aligned mean per joint position error (PA-MPJPE) and Procrustes-aligned mean per vertex position error (PA-MPVPE) refer to the MPJPE and MPVPE after aligning the predicted hand results with the Ground Truth using Procrustes alignment, respectively. To better investigate the impact of incorporating additional 2D information on performance, we introduce MPJPE-XY, MPJPE-Z, MPVPE-XY, and MPVPE-Z in the ablation study. These metrics calculate the hand recovery error of MPJPE and MPVPE relative to the ground truth in the XY and Z dimensions, respectively.

## 4.4 Comparison with state-of-the-art methods

**Quantitative results in InterHand 2.6M datasets.** We conduct a comprehensive comparison of our method with recent state-of-the-art (SOTA) hand pose and shape estimation methods on the InterHand 2.6M test dataset, as presented in Table 1. The MRRPE metric effectively reflects the estimation performance of relative hand distances. Our method achieves the best performance on this metric with 21.80mm, surpassing InterWild, Ren et al., and 4DHands by 4.94mm, 7.18mm, and 2.78mm respectively. Our method also demonstrates consistent improvement in MPJPE and MPVPE, outperforming the current best method, 4DHands, by 2.08mm and 2.05mm respectively. Furthermore, we observe consistent performance gains in both the IH MPJPE/MPVPE and SH MPJPE/MPVPE metrics, highlighting the generalizability and robustness of our method for both single-hand and interacting hand estimation.

**Quantitative results in HIC.** We present the results on the HIC dataset (Tzionas et al., 2016), which features in-the-wild cross-hand data, to evaluate performance in real-world scenarios. In Table 2, we compare these results with IntagHand, InterWild, and 4DHands, a state-of-the-art method specifically designed for two-hand recovery in the wild. The training sets for these models don't contain the HIC dataset. In comparison to the MRRPE metric, our method achieved improvements of 2.92mm and 4.09mm over 4Dhands and InterWild, respectively. In comparison to the MPJPE metric, our method showed improvements of 2.96mm and

Table 2: Comparison with state-of-the-art on HIC dataset (Tzionas et al., 2016).

| Methods | MRRPE | MPJPE | MPVPE |
|---|---|---|---|
| IntagHand (Li et al., 2022) | 73.04 | 20.38 | 21.56 |
| InterWild (Moon, 2023) | 26.43 | 15.62 | 15.17 |
| 4DHands (Lin et al., 2024) | 25.26 | 9.32 | 9.93 |
| **Ours** | **22.34** | **6.36** | **6.62** |

9.26mm compared to 4Dhands and InterWild, respectively. Lastly, in comparison to the MPVPE metric, our method led to enhancements of 3.31mm and 8.55mm over 4Dhands and InterWild, respectively. This demonstrates that our method has greater stability on unseen data.

**Quantitative results with single-hand method on FreiHAND dataset (Zimmermann et al., 2019).** We also compared our method with the latest state-of-the-art transformer-based single-hand recovery methods on the Frei-HAND dataset (Zimmermann et al., 2019). In this comparison, our method omits the estimation of the relative translation between the two hands and the execution of diffusion denoising for both hands. The comparison with single-hand methods demonstrates the effectiveness and robustness of integrating multiple 2D inputs for hand recovery. As shown in Table 3, compared to HaMer and Zhou et al., we

Table 3: Comparison with state-of-the-art on Frei-HAND dataset (Zimmermann et al., 2019).

| Methods | PA-MPJPE | PA-MPVPE |
|---|---|---|
| METRO (Zhang et al., 2019) | 6.7 | 6.8 |
| MeshGraphomer (Lin et al., 2021) | 6.3 | 6.5 |
| AMVUR (Jiang et al., 2023) | 6.2 | 6.1 |
| HaMeR (Pavlakos et al., 2024) | 6.0 | 5.7 |
| Zhou et al. (Zhou et al., 2024) | 5.8 | 6.1 |
| **Ours** | **5.1** | **5.2** |

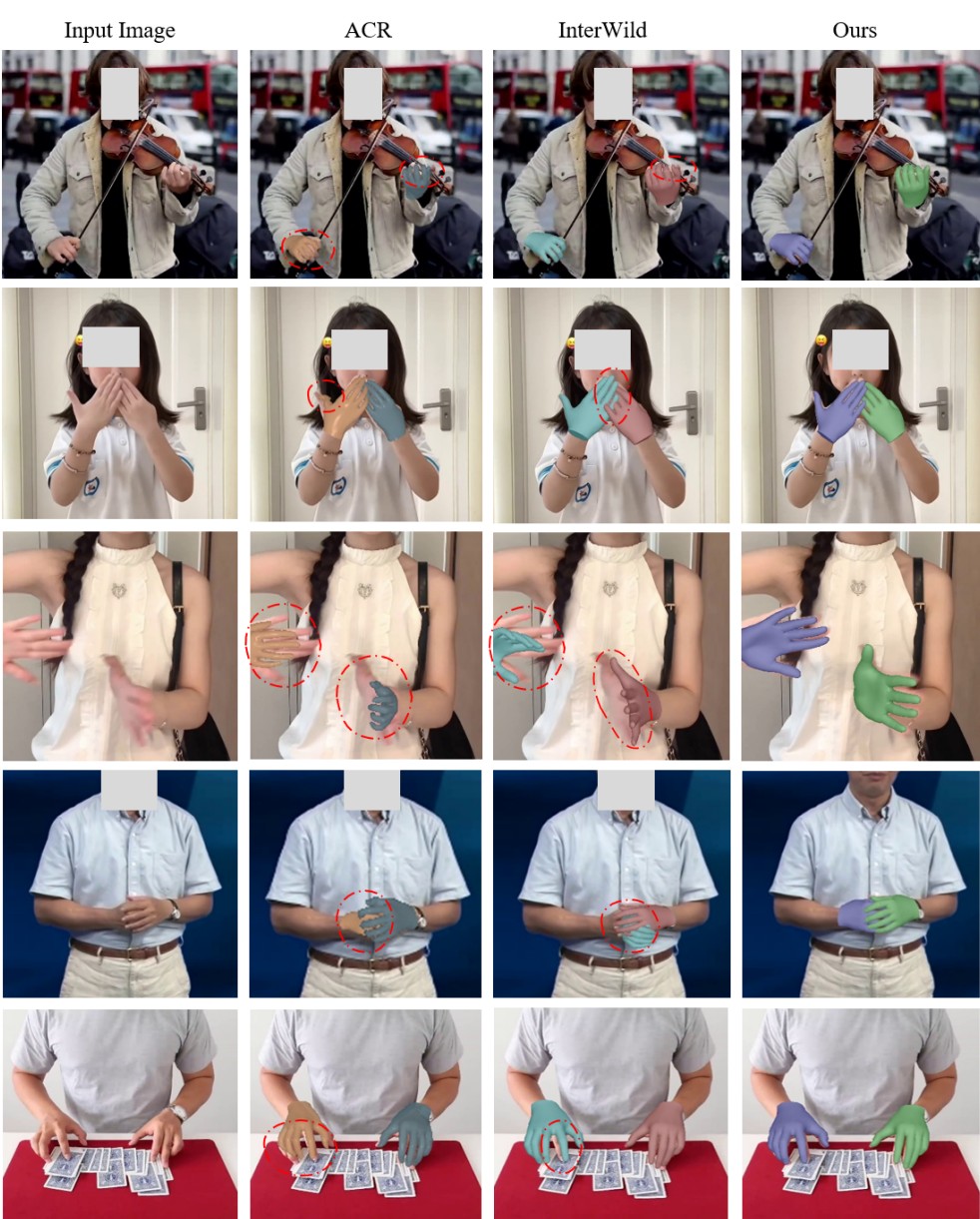

Figure 3: Qualitative results in real scenes. The images are all sourced from the internet. The red circle indicates distortion or inaccurate estimation.

achieved improvements of 0.8mm and 0.4mm
in PA-MPJPE, and 0.6mm and 0.8mm in PA-MPVPE, respectively. These results validate the general applicability of our approach in incorporating additional information for single-hand estimation.

**Qualitative results.** We present a visual comparison against the open-sourced ACR and InterWild methods using real-world images. Performance on real-world data can better highlight the differences between methods. As shown in Fig. 3, we compare our method with the open-sourced ACR and InterWild on real-world images sourced from the internet. In the first row, both ACR and InterWild exhibit misalignment issues in their estimations. In the second row, while ACR successfully estimates the relative distance between the two hands, it suffers from thumb distortions. InterWild, on the other hand, shows hand penetration issues. The third row presents an image with partial occlusion and blur affecting the hands. Both ACR and InterWild fail to recover the hand poses ac-

Table 4: Ablation studies on InterHand2.6M (Moon et al., 2020).

| Methods | MRRPE | MPJPE | MPVPE | MPJPE-XY | MPJPE-Z | MPVPE-XY | MPVPE-Z |
|---|---|---|---|---|---|---|---|
| Baseline | 25.30 | 7.77 | 7.93 | 5.21 | 4.54 | 5.29 | 4.63 |
| + 2d Keypoints | 24.40 | 6.43 | 6.67 | 4.22 | 4.36 | 4.35 | 4.45 |
| + Segmentation map | 24.20 | 6.14 | 6.34 | 4.12 | 4.32 | 4.22 | 4.41 |
| + Depth map | 21.93 | 5.63 | 5.92 | 4.09 | 3.31 | 4.15 | 3.40 |
| + Two-hand Diffusion model | **21.80** | **5.41** | **5.67** | **4.02** | **3.24** | **4.09** | **3.34** |

curately, whereas our method achieves a relatively successful estimation. In the fourth row, ACR generates an incorrect estimation for the left hand, while InterWild again exhibits hand penetration problems. Finally, in the last row, both ACR and InterWild display misaligned hand estimations.

## 4.5 ABLATION STUDY

The main contributions of our method lie in the introduction of additional 2D information for two-hand pose estimation and the proposal of a two-hand refinement module based on a diffusion model. To validate the effectiveness of these contributions, we conducted comprehensive ablation studies on the InterHand 2.6M dataset in Table 4. We also additionally report the MPJPE/MPVPE-XY and MPJPE/MPVPE-Z metricswhich can effectively demonstrate the improvements contributed by different types of information to the model's performance.

**Effectiveness of different information inputs.** As shown in Table 4, we gradually added different types of information for fusion to observe their impact on performance. We observed that adding 2D keypoints and segmentation maps resulted in significant improvements in MPJPE and MPVPE, with the greatest reduction in the XY dimension estimation error. It is easy to understand that 2D key-points and segmentation maps provide excellent XY dimension information cues. Furthermore, we found an interesting phenomenon that the improvement from incorporating 2D keypoints was greater than that from segmentation maps. This is because 2D keypoints provide more fine-grained 2D in-formation about each joint of the hand, while segmentation maps only provide contour information. Moreover, after fusing depth maps, we observed significant improvements in MPJPE/MPVPE-Z and MRRPE. Depth maps can clearly help the model better reason about the hierarchical relationships of the 3D hand.

**Effectiveness of using a two-hand diffusion model.** Table 4 also demonstrates the impact of the two-hand diffusion model on hand recovery performance. We can see that after adding diffusion, MRRPE, MPJPE, and MPVPE all achieve improvements, with increases of 0.13 mm, 0.22 mm, and 0.25 mm, respectively, and with the same improvement trend in both the XY and Z dimensions. The two-hand diffusion model can effectively learn the reasonable distribution of two-hand data, which can help adjust the relative relationship between the hands to a more reasonable state.

## 5 CONCLUSION

In this paper, we propose a two-hand reconstruction method that integrates additional 2D reference information to improve hand alignment and depth recovery performance. Furthermore, when one hand is occluded by another (making the 2D reference information for the occluded hand unreliable), we introduce the use of a two-hand diffusion model as a prior to address the penetration issue. Extensive qualitative and quantitative experimental results demonstrate that our method significantly outperforms previous two-hand and single-hand reconstruction approaches. **Limitation and Future Work:** One limitation of this work is that the inference speed may be slower compared to the direct estimation of hand parameters. This is because our approach requires running an additional foundation model to infer 2D information input. Although the introduction of two-hand diffusion may further reduce inference speed, we have implemented a hand penetration detection mechanism that can filter out unnecessary denoising diffusion processes. We believe that with the increasing efficiency of base models and advances in GPU hardware, this issue of inference speed can be mitigated. Additionally, our method still faces challenges in handling extreme occlusions and motion blur in hand images, as the additional 2D information may become unreliable under such conditions. We believe that future integration of temporal processing could effectively alleviate this problem.

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
