# OpenReview forum: "Robust Two-Hand Reconstruction with Additional 2D Information and Diffusion Prior"
_ICLR.cc/2025/Conference — ICLR 2025 Conference Withdrawn Submission_

### Official Review · Reviewer_R2Xr · 2024-10-17

**Soundness:** 3
**Presentation:** 3
**Contribution:** 2
**Rating:** 5
**Confidence:** 4

**Summary:**

The paper presents a novel approach that utilizes three key components: 1) 2D keypoints, 2) segmentation maps, and 3) depth maps derived from advanced foundation models to facilitate the reconstruction of two hands. Additionally, the authors introduce a two-hand diffusion model specifically designed to refine instances of hand penetration. Experimental results demonstrate that the substantial information provided by foundation models, coupled with training on large-scale datasets, contributes to achieving robust reconstruction performance on publicly available benchmarks.

**Strengths:**

1. The overall performance is robust. Both qualitative and quantitative results are better than current approaches.
2. The paper is easy to follow.

**Weaknesses:**

1. The comparisons with existing works lack fairness, which obscures the clarity of performance improvements. For instance:
   1. On the InterHand2.6M dataset, certain models (e.g., *Ren et al. [1]*, *IntagHand [2]*) are exclusively trained on InterHand2.6M, while *InterWild* ensures fair comparisons by training all baselines on the same dataset.
   2. Similarly, on the FreiHAND dataset, some models (e.g., *HaMeR [3]*, *METRO [4]*) are trained solely on FreiHAND.

2. There is a notable lack of discussion regarding optimal utilization of information from the powerful foundation model (Sapiens). While employing auxiliary information to enhance performance is a relatively straightforward approach, the paper would benefit from providing deeper insights into how the community can effectively leverage this information. As it stands, the performance improvements appear primarily attributable to the inherent strengths of Sapiens. It is also unclear whether a weaker intermediate representation would lead to a significant decline in performance.

3. The effectiveness of the diffusion model is not adequately validated:
   1. The performance improvements associated with the diffusion model are minimal in quantitative evaluations, particularly when compared to the enhancements derived from the foundation model.
   2. There are no visual results demonstrating that the diffusion model effectively addresses the issue of hand penetration.
   3. The rationale for utilizing the diffusion model remains ambiguous. Is it demonstrably superior to alternative approaches?


[1] Ren, Pengfei, et al. "Decoupled iterative refinement framework for interacting hands reconstruction from a single rgb image." Proceedings of the IEEE/CVF International Conference on Computer Vision. 2023.

[2] Li, Mengcheng, et al. "Interacting attention graph for single image two-hand reconstruction." Proceedings of the IEEE/CVF Conference on Computer Vision and Pattern Recognition. 2022.

[3] Pavlakos, Georgios, et al. "Reconstructing hands in 3d with transformers." Proceedings of the IEEE/CVF Conference on Computer Vision and Pattern Recognition. 2024.

[4] Lin, Kevin, Lijuan Wang, and Zicheng Liu. "End-to-end human pose and mesh reconstruction with transformers." Proceedings of the IEEE/CVF conference on computer vision and pattern recognition. 2021.

**Questions:**

1. Is it possible to fine-tune the foundation model to improve the performance?
2. Are there any failure cases?

---

### Official Review · Reviewer_FaZR · 2024-10-19

**Soundness:** 2
**Presentation:** 3
**Contribution:** 2
**Rating:** 3
**Confidence:** 5

**Summary:**

This paper presents a 3D interacting hand reconstruction system. Unlike existing works that utilizes implicit feature maps to regress 3D hand parameters, the proposed one estimates explicit geometric features, such as 2D keyepoints, segmentation, and depth maps, and uses them as an intermediate representation to regress 3D hand parameters. In addition, diffusion-based prior is employed to prevent collision between two hands. Strong experimental results demonstrate the effectiveness of the proposed system.

**Strengths:**

It is quite easy to follow the manuscript. The system is concise and simple.

**Weaknesses:**

1. Novelty is not enough. In other words, there are not many new things. Combining geometric features, such as 2D keypoints and segmentations, have been tried in a number of previous works. For example, Pos2Mesh (ECCV 2020) and 3DCrowdNet (CVPR 2022) used off-the-shelf 2D keypoint detectors. Utilizing geometric features to enhance the performance have been tried many times, so this should not be a novelty, while the authors argue that this is one of the major novel contributions.

2. Running time. Sapiens is used to get the geometric features, and DDIM is used for the diffusion-based collision handling. Both should take a long time. Sapiens, despite its strong accuracy, is slow as it takes 1K resolution images. DDIM, due to its iterative nature, is slow. This is discussed in the limitation section (L478).

3. Lack of interesting demos and qualitative results. The submission does not have supplementary material and video demos. Given the strong performance of the proposed method, I expected a number of impressive video demos. Unfortunately, they are not available.

**Questions:**

1. How did the authors normalize the depth maps from Sapiens? The depth map from it has scale and translation ambiguity as the input is a single image.
2. The collision detection (Eq. 3) is a little bit weird. Colliding vertices could have any dot products. For example, if the right and left hands are overlapped with the same wrist position, then the dot product should be close to 1. Also, when hands are seeing each other from the opposite direction, the dot product should be close to -1.
3. How the diffusion-based collision detector works compared to existing collision solvers, such as SDF-based ones (Monocular 3D Reconstruction of Interacting Hands via Collision-Aware Factorized Refinements. 3DV 2021)?

---

### Official Review · Reviewer_DkyM · 2024-11-02

**Soundness:** 2
**Presentation:** 2
**Contribution:** 1
**Rating:** 3
**Confidence:** 4

**Summary:**

This manuscript proposes a method for the recovery of 3D meshes of two (possibly) interacting hands. In addition to using a ResNet-50 based model to predict MANO parameters from cropped images of hands, the authors proposed to take all of the outputs from the Sapiens foundation model (keypoints, segmentation, depth maps) and use all features derived from these predictions as input to a transformer encoder to predict the MANO parameters of two hands. To handle the edge-case of almost fully-overlapping hands, they also use a pre-trained denoising diffusion model to refine the initial mesh predictions. The overall approach achieves state-of-the-art on InterHand2.6M, HIC, and FreiHAND datasets on all evaluated metrics (MRRPE, MPJPE, MPVPE, etc.)

**Strengths:**

The manuscript has made reasonable efforts to explain the proposed method and cites relevant work (though most cited works are from 2022 onwards). Some effort has been made to motivate the use of the many additional features (taken from Sapiens) during hand shape recovery. The ablation study in Tab. 4 is intuitive (incremental improvements with depth map being most helpful), and the main comparisons against state-of-the-art seem to show overwhelmingly positive results.

**Weaknesses:**

This submission seems to throw “everything and the kitchen sink” at the problem of hand shape recovery. Each of the 3 Sapiens models used (separate models required for keypoint, segmentation, and depth) contains a minimum of 300 million parameters and can each go up to 2 billion parameters (the authors do not specify which model they chose). As a post-processing step, the authors also apply InterHandGen without any modification to its weights. I wonder if it is really surprising that such a heavy-handed approach results in performance improvements compared to the methods they compare to. The authors identify the ludicrosity of their own work by mentioning that their “inference speed may be slower” - which seems understated.

The proposed solution is certainly a respectable engineering effort. For settings that can afford the inference requirements of the proposed solution, the manuscript’s insights could be valuable. However, there is no other insight provided by the paper. The ablation study simply turns each of the external models’ contributions on one-by-one and is hardly surprising. The authors put some effort into explaining that they apply the diffusion model to refine the shape of the non-occluded hand first. However, this and other design decisions are not explained using either quantitative or qualitative results.

**Questions:**

- What is the complexity of your model in comparison to the state-of-the-art (# params, FLOPs, or MACs)?
- What are the implementation details of your architecture? E.g. transformer parameters, architecture of the hand regressor module, information about the diffusion architecture.
- Did you perform any ablation studies to validate your many other design decisions?
- How exactly are the inputs provided to the transformer encoder? Your Eq. 1 implies that all features are concatenated, but that may yield very few tokens. Which tokens are defined and how?

---

### Official Review · Reviewer_xy6J · 2024-11-03

**Soundness:** 3
**Presentation:** 3
**Contribution:** 2
**Rating:** 3
**Confidence:** 4

**Summary:**

This paper presents a sound approach for estimating 3D hand pose and shape for dual hands from monocular RGB images by leveraging multiple foundational models. Specifically, it incorporates 2D keypoint detection, segmentation, depth, and 2D feature maps as supplementary information to enhance estimation accuracy beyond that achievable with RGB images alone. The authors address a common challenge in two-hand pose estimation—significant hand overlap leading to interpenetration—by employing a cascaded denoising diffusion model. This model iteratively refines hand positions, using collision loss and gradient guidance to correct occlusions and ensure realistic hand interactions. Experimental results demonstrate that this method surpasses current benchmarks on InterHand2.6M, HIC, and FreiHAND, highlighting its effectiveness in handling complex two-hand poses and occlusions.

**Strengths:**

This paper leverages a diffusion prior method to address certain limitations in previous approaches, with an intuitive strategy of conditioning on the visible hand. Unlike prior methods that treat penetration refinement as a test-time adaptation, this approach achieves end-to-end processing. I appreciate the detailed experiments and comparisons presented in the paper. The results effectively demonstrate the model’s robust hand reconstruction capabilities, showcasing its potential in handling complex hand poses and occlusions. In term of clarity: The paper’s motivation is clear and straightforward—using more foundational model information to enhance the current model's performance. The writing is direct, making the paper easy to read and follow. In term of significance, However, for applications in AR, practicality is essential. The heavy reliance on multiple foundational models could hinder real-time applicability. Additionally, faster, more efficient methods exist for addressing interpenetration issues, such as using primitive collision shapes as proxies. The denoising diffusion approach may appear overly complex for this purpose.

**Weaknesses:**

A major concern is that adding additional information to enhance vision tasks is already a common practice. Many existing works assume that segmentation, depth, and other foundational model outputs are accessible during both training and inference. Therefore, this strategy should not be considered a vital contribution of the paper. It’s reasonable to assume that any contemporary model, given the outputs of foundational models during training, could achieve comparable results to the proposed approach.
Regarding the second contribution, although the paper uses a diffusion model to mitigate interpenetration issues, it lacks experimental validation for this claim. There is a noticeable absence of detailed experiments demonstrating the effectiveness of this approach, as well as comparisons with other methods for handling interpenetration.
For example, it would be useful to quantify improvements by showing reductions in penetration volume or depth, or the percentage of vertices with reduced penetration. Additionally, it remains unclear whether the diffusion model, while denoising to reduce interpenetration, might compromise the accuracy of the original pose estimation. Another issue is that if the ground truth (GT) in the dataset inherently includes interpenetration, comparing results against this GT versus a “non-penetrative” output might lack meaningful impact. Moreover, the paper does not compare its method with other common post-processing (test-time adaptation, or TTA) approaches, such as fitting-based techniques like GraspTTA, ContactOpt, or CPF.
Overall, while I acknowledge the strong engineering effort and promising results of this work, the weaknesses are also evident. As the title suggests, "WITH ADDITIONAL 2D INFORMATION AND DIFFUSION PRIOR," the first component should not be considered an original contribution (given the abundance of similar strategies in existing work), and the second lacks crucial experimental validation.

**Questions:**

Would it be helpful to quantify improvements by showing reductions in penetration volume or depth, or the percentage of vertices with reduced penetration? Additionally, is it clear whether the diffusion model, while denoising to reduce interpenetration, might compromise the accuracy of the original pose estimation? If the ground truth (GT) in the dataset inherently includes interpenetration, does comparing results against this GT versus a “non-penetrative” output have meaningful impact? Finally, why does the paper not compare its method with other common post-processing (test-time adaptation, or TTA) approaches, such as fitting-based techniques.

---

### Note · Authors · 2024-11-15

I have read and agree with the venue's withdrawal policy on behalf of myself and my co-authors.